# Hydrogel Conjugation: Engineering of Hydrogels for Drug Delivery

**DOI:** 10.3390/pharmaceutics17070897

**Published:** 2025-07-10

**Authors:** Linh Dinh, Sung-Joo Hwang, Bingfang Yan

**Affiliations:** 1Division of Pharmaceutical Sciences, James L. Winkle College of Pharmacy, University of Cincinnati, Cincinnati, OH 45229, USA; yanbg@ucmail.uc.edu; 2College of Pharmacy & Yonsei Institute of Pharmaceutical Sciences, Yonsei University, 85 Songdogwahakro, Yeonsu-gu, Incheon 21983, Republic of Korea; sjh11@yonsei.ac.kr

**Keywords:** hydrogels, hydrogel conjugation, wound care, drug delivery systems, tissue engineering

## Abstract

**Background**: Hydrogels are 3D networks of hydrophilic polymers with various biomedical applications, including tissue regeneration, wound healing, and localized drug delivery. Hydrogel conjugation links therapeutic agents to a hydrogel network, creating a delivery system with adjustable and flexible hydrogel properties and drug activity, allowing for controlled release and enhanced drug stability. Conjugating therapeutic agents to hydrogels provides innovative delivery formats, including injectable and sprayable dosage forms, which facilitate localized and long-lasting delivery. This approach enables non-viral therapeutic methods, such as insertional mutagenesis, and minimally invasive drug administration. **Scope and Objectives**: While numerous reviews have analyzed advancements in hydrogel synthesis, characterization, properties, and hydrogels as a drug delivery vehicle, this review focuses on hydrogel conjugation, which enables the precise functionalization of hydrogels with small molecules and macromolecules. Subsequently, a description and discussion of several bio-conjugated hydrogel systems, as well as binding motifs (e.g., “click” chemistry, functional group coupling, enzymatic ligation, etc.) and their potential for clinical translation, are provided. In addition, the integration of therapeutic agents with nucleic acid-based hydrogels can be leveraged for sequence-specific binding, representing a leap forward in biomaterials. **Key findings**: Special attention was given to the latest conjugation approaches and binding motifs that are useful for designing hydrogel-based drug delivery systems. The review systematically categorizes hydrogel conjugates for drug delivery, focusing on conjugating hydrogels with major classes of therapeutic agents, including small-molecule drugs, nucleic acids, proteins, etc., each with distinct conjugation challenges. The design principles were discussed along with their properties and drug release profiles. Finally, future opportunities and current limitations of conjugated hydrogel systems are addressed.

## 1. Introduction

Hydrogels, 3D networks of cross-linked hydrophilic polymers, are a class of materials with unique properties that have been used in various applications. Hydrogels are formed by dissolving hygroscopic polymeric materials in water. Due to their hygroscopic nature, hydrogels can hold large amounts of fluid in the swelling stage. In clinical practice, hydrogels are commonly used as an absorbent. They function as scaffolds for tissue growth, wound covering films, and physical barriers to block or minimize adhesions. In biomedicine, hydrogels have been extensively utilized in tissue regeneration, implants, physiological and pathological mechanism studies, bioimaging, biosensors, biodevices, and therapeutic drug delivery. The hydrogel market was valued at USD 27.72 Billion in 2024 and is projected to grow to USD 59.85 Billion by 2034 with both drug-free hydrogels (e.g., Vagisil^®^ [1], Viscotears^®^ [2], ACUVUE^®^ [3], Guardix^®^-SG [4], GranuGEL^®^ [5], etc.) and hydrogel drug delivery products (Gaviscon^®^ [6], OncoGel^®^ [6,7], Ozurdex^®^ implant [6,8], etc.).

In pharmaceutics, the physicochemical properties of hydrogels are typically engineered according to the Quality Target Product Profile. For example, covalent cross-linking with a high cross-linking density results in hydrogels with a high mechanical strength, dense structure, and stiffness [9]. In contrast, weak physical cross-linking, a lower cross-linking degree, and a lower concentration of cross-linking agents can create a loose hydrogel network, thus increasing the flowability of the hydrogel [9,10]. In addition, the various sources of hydrogel materials, their variable compositions, as well as their fabrication approaches correspond to the porous structures, physicochemical, and mechanical characteristics of the hydrogels, which determine their applications [10]. A major consideration in creating a hydrogel system is to employ suitable polymers. Natural and naturally derived polymers are preferred due to their low toxicity, biocompatibility, and biodegradability. Many synthetic polymeric materials with well-defined structures and longer shelf life can be designed to tailor the sol-to-gel state, degradability, and functionality [11]. Multi-component hydrogel systems have been employed in cell culture hydrogel studies, promoting cell expansion, differentiation, organization, and tissue engineering. Two or more polymers incorporated in a system allow the hybrid polymer to have a wide range of physicochemical and biological properties [12,13]. Furthermore, hydrogels can be engineered via numerous polymerization and chemical and physical cross-linking methods to undergo in situ gelation for injectability. Generally, a hydrogel product is heavily influenced by its material composition and concentration, preparation methods, cross-linking formation, physical, chemical, and biochemical properties, as well as its physical response. Thus, the classification of hydrogels depends on their properties (such as composition, network electrical charge, configuration, structure, durability, nature of swelling…), fabrication approaches, and origins (Figure 1). Figure 1 provides the foundational framework for the development of hydrogel systems for biomedical applications. Figure 2 includes schematic images for step procedures involving conjugated hydrogel synthesis, uncovering the influences of hydrogel conjugation on their biological and therapeutic activities and applications. Figure 2 highlights how the conjugation strategies influence the classification of hydrogel systems into (A) drug-free hydrogel and (B) drug-loaded hydrogel, which are outlined in Figure 1, and in turn, affect the application and therapeutic efficacy of the resulting hydrogel systems.

This review focuses on the hydrogel conjugation process with therapeutic molecules and conjugation approaches for hydrogel-based disease therapies. Biodegradability and long-term stability play important roles in the design and application of hydrogel-loaded drug systems. The system degradation mechanisms and hydrogel degradation rates are influenced by the hydrogel’s composition, the stimuli and environmental conditions, and the loaded drug. The synthesis, characterization, and physicochemical properties of hydrogels and hydrogels as a delivery vehicle have been extensively covered elsewhere [9,10,11,12,14]. The review systematically categorizes hydrogel-conjugates for drug delivery, focusing on conjugating hydrogels with major classes of therapeutic agents.

Linking therapeutic agents to a hydrogel network combines the tunable and flexible hydrogel properties and therapeutic activity to utilize hydrogels as drug delivery systems. As shown in Figure 2, hydrogel conjugation for drug delivery includes binding or chemically linking the drug molecules to a hydrogel network. A drug remains “trapped” within the gel structure, reaches the therapeutic target, and is eventually released in a controlled manner. Similarly, a cell-loaded hydrogel system can be considered a bio-conjugated hydrogel system for cell-based therapies. The hydrogel 3D network can facilitate the delivery of cells to targeted sites of action to promote cell growth and integration [12]. Interestingly, DNAs and RNAs, genetic information stored in natural biopolymer chains, can be conjugated to hydrogels to facilitate their delivery for use in gene therapy (Figure 2). Nucleic acids can be engineered to conjugate to the polymeric backbone of the hydrogel through covalent conjugation, resulting in more stable nucleic acid-hydrogel complexes that can sustain the release of the therapeutics, potentially enhancing in vivo efficacy with minimal toxicity [15]. This review focuses on drug-loaded hydrogel systems and drug–hydrogel conjugation strategies, highlighting the conjugation between the hydrogel and the drug. Hydrogel conjugation is defined as attaching a chemical drug or a biomolecule to a hydrogel. Table 1 classifies hydrogel–drug conjugates based on their fundamental components—the hydrogel component and the drug component. In addition to considering the properties of hydrogel materials (Figure 1), the drug characteristics must be considered when determining conjugation strategies.

## 2. Conjugation of Small Molecules to Hydrogels

Small-molecule drugs comprise 90% of the pharmaceutical market and have continued to lead in approved therapies and new drug approvals. The prevalent use of small molecules–hydrogel conjugates makes them a classic approach in wound healing and drug delivery. Small-molecule drugs are typically loaded into hydrogels by physical entrapment. Hydrophilic drugs are often dissolved in hydrogels, and hydrophobic drugs can be dissolved or dispersed in the hydrogel or a pre-gel solution with the help of additives.

### 2.1. Hydrogels in Wound-Healing and Drug Delivery

Hydrogels have been used as integral components in wound dressings because they are water-swellable polymer networks that can mimic the inherent characteristics of the natural extracellular matrix. Hydrogels can cool burns and absorb exudates in wounded areas (Figure 3). Moreover, owing to their tunable and flexible physical properties, hydrogels are ideal materials not only in tissue engineering and neural regeneration but also in drug delivery, where hydrogels are used as delivery vehicles for various therapeutic agents as depicted in Figure 2. Particularly, for small molecules, the high-water content of hydrogels plays an important role in their incorporation. Drug release from hydrogels is governed by diffusion and occurs simultaneously with hydrogel erosion. Because hydrogel erosion and drug release studies cannot be conducted simultaneously, the common practice is to perform the two experiments separately, then make connections between the release profiles and erosion rates. “Burst release” is a widely reported and recognized phenomenon in hydrogel drug delivery systems. It poses a challenge because it can affect the pharmacokinetic profile of the loaded drug; there is no mechanistic theory put forth or solution for prevention [16]. Typically, through different administration routes involving oral, buccal, injectable, vaginal, ocular, and transdermal, hydrogels serve as a local “carrier” that can leverage the therapeutic outcomes of the drug encapsulated by delivering high drug concentrations to a target site in a controlled manner via stimuli-responsive strategies (Figure 3).

Several hydrogels used as drug delivery systems are presented in Table 2, highlighting the hydrogel constituents, active pharmaceutical ingredients (small-molecule drugs), methods of conjugation, administration route, stimuli-responsiveness, and drug release pattern. Wound dressing and transdermal delivery are the most common hydrogel applications [17]. The skin is the largest external organ and provides a great administration route for high patient compliance and controlled-release formulations [18]. Small molecules, under 500 Da, can pass the stratum corneum [19]. From patches to medical devices, transdermal delivery represents a revolutionary alternative to oral delivery, as hydrogels used as transdermal delivery systems bypass first-pass metabolism. Hydrogels can hydrate and enhance skin elasticity, thus facilitating skin moisturization and enhancing skin permeation, allowing active ingredients to penetrate more effectively. Additionally, many hydrogels can be designed to exhibit mucoadhesiveness to adhere to and penetrate the mucosal surface [20].

### 2.2. Hydrogels for Targeted Delivery

A variety of mucoadhesive biopolymers including carbomers, chitosan, gelatin, hyaluronic acid, lectins, sodium alginate, and cellulose-based are commonly used to deliver drugs to mucosal membranes, such as the eyes, nose, and vagina (Table 2). For local delivery, mucoadhesive hydrogel systems that can control drug release rates in the vaginal canal and extend residence time, are currently being utilized. Commercially, hydrogel-based mucoadhesive patches and tablets are ideal for buccal administration and local drug delivery. The ease of administration and the fact that buccal delivery can avoid the gastrointestinal tract and first-pass metabolism effect are the advantages of mouthcare products such as Biotène^®^ and SCHALI^®^ Dental Care Hydrogel [6,21,22,23,24]. Interestingly, functional toothpaste has recently been manufactured to expand the oral care business and cross paths with pharmaceutical ingredient delivery [24,25,26]. Furthermore, hydrogel smart oral delivery systems (pH-responsive hydrogels) offer targeted release of drugs within the complex gastrointestinal tract, as the oral route remains a popular drug delivery method. In recent years, despite systemic delivery challenges, injectable hydrogels have been increasingly favored over conventional gel formulations due to their minimally invasive application, resulting in less patient discomfort and on-demand shape-forming adaptability for difficult-to-access tissues. The mucoadhesiveness ensures small-molecule drugs remain at the site of action and delivers the drug to targeted tissues.

### 2.3. Stimuli-Responvise Hydrogels

Initially, hydrogels were composed of a single polymer. Most commercially available hydrogel-based drug delivery products are single-polymer systems (Table 2), primarily because they offer predictable degradation, regulatory approval ease, and well-established safety profiles. However, most monomeric systems had limited control over drug release. Advancements in tailoring degradation rates, responsive behaviors, and structural modifications enable controlled and targeted drug release. Copolymeric systems allow diverse applications in response to different physiological conditions (Table 2). For example, poly (lactic-co-glycolic acid) (PLGA), which degrades through hydrolytic cleavage of its ester bonds in the backbone of lactic acid (PLA) and glycolic acid (PGA) units, has been widely used for injectable hydrogel systems that gradually release drugs over time. PLGA with higher PGA content leads to faster degradation due to its higher hydrophilicity, while PLGA with a higher PLA content is more hydrophobic and degrades more slowly [27]. Stimuli-responsive hydrogels have further expanded hydrogel applications by incorporating polymers that react to external triggers. Due to the cationic nature of amino groups, chitosan exhibits pH-sensitive swelling behavior, allowing targeted drug release in acidic environments such as the stomach or infected tissues [28]. Temperature-sensitive polymers like poloxamers or poly [N-isopropyl acrylamide) (NIPAM)] undergo phase transitions at body temperature, enabling on-demand drug release [4,29,30]. The application of Boolean logic principles (“AND”, “OR”, “XOR”, and “NOT” gates) in hydrogel drug delivery systems that respond to specific biological signals, allows highly targeted drug release, maximizing therapeutic effects and minimizing off-target drug exposure [31]. The smart hydrogels are engineered with responsive materials that act as “gates”, for example, PNIPAM and poly acrylic acid can respond to temperature and pH inputs because PNIPAM shrinks above its lower critical solution temperature [26,27,28], and poly acrylic acid swells at pH above its pKa [32]. Consequently, a blend of PNIPAM and polyacrylic acid can implement an “AND gate” allowing drug release only when both the surrounding environmental temperature and pH are above thresholds (Figure 4).

**Table 2 pharmaceutics-17-00897-t002:** Examples of hydrogels as drug delivery systems.

	Hydrogel Constituents	Drugs	Method of Conjugation *	Release	**Administration Routes**	**Stimuli-Responsive**
Zilactin-B Gel^®^ by Zila Pharmaceuticals (Phoenix, AZ, USA) [17]	(Hydroxypropyl)methylcellulose (HPMC)	Benzocaine	drug dispersed in gel solution	4–6 h pain-free on the oral mucosa	Buccal	
Thermosensitive poloxamer 407-based gel for ultrasound-mediated inner ear drug delivery [29]	Albumin-shelled microbubble gel, poloxamer	dexamethasone		sustained release of dexamethasone from the middle ear. On post-treatment days 1 and 7, the treated groups showed significantly higher drug levels than the injection group.	intratympanic	temperature
Kaletra^®^ film-coated tablets by AbbVie Ltd. (Chicago, IL, USA) [33]	Polyvinyl alcohol (PVA)	Lopinavir/ritonavir		Obtained steady-state pharmacokinetic properties and parameters of lopinavir	Oral	
pH-sensitive hydrogel films for oral administration [34]	sodium tripolyphosphate cross-linked ternary blended chitosan, guar gum, polyvinylpyrrolidone	Ciprofloxacin hydrochloride	drug dissolved in aqueous gel solution	30% of the drug was released in the first 30 min in simulated gastric fluid, and sustained release was observed in simulated intestinal fluid and phosphate-buffered saline	Oral ^1^	pH
pH-sensitive hemicellulose/graphene oxide-based hydrogel for oral administration [35]	Physically cross-linked hemicellulose and graphene oxide	Vitamin B12		Controlled intestinal release and decreased ineffective stomach release of the drug.	Oral	pH
Drinkable liquid in situ-forming tough (LIFT) hydrogels [36]	Chemically cross-linked alginate, four-arm-PEG-maleimide	Lumefantrine	drug dispersed in gel solution	Hydrogel formulations resulted in peak plasma drug concentrations at 24 h, meanwhile free drug resulted in peak plasma concentrations at 5–7 h post-administration. Hydrogels can deliver comparable total drug doses as free drug at lower plasma concentrations.	Oral	pH
Astero^®^ by Gensco Pharma (Doral, FL, USA) [37]	Polyethylene glycol (PEG)	Lidocaine hydrochloride	drug dissolved in aqueous gel solution	Fast pain relief with an onset of action within 3–5 min.	Transdermal	
Electro-responsive conductive hydrogel patch [38]	photo cross-linked gelatin methacrylate, alginate, and silver nanowire	Doxorubicin	drug dissolved or dispersed in aqueous gel solution; cationic doxorubicin interacts with hydrogel anionic groups ^2^	Hydrogel patch controlled on-off drug release. Controlled drug release through hydrogel osmotic pressure and structural changes by electrical stimulation.	Transdermal	Electrical stimulation
Drug loaded- magnetite nanoparticles dispersed in hydrogel beads [39]	PNIPAM, methylene bisacrylamide, sodium alginate	Dexamethasone	drug dissolved or dispersed in the aqueous pre-gel solution	Light exposure enables on-demand drug release. Cumulative drug release was 24% for the first 10 h and 50% in 40 h, then reaching an equilibrium value of 66% in 120 h. The release rate could be adjusted by light intensity.	Transdermal	Light
Encare^®^ by Blairex Laboratories Inc. (Columbus, IN, USA) [40,41]	PEG	Nonoxynol-9	drug dissolved in aqueous gel solution	1 suppository should be inserted at least 10 min before intercourse and provide effective contraception for up to 1 h after insertion.	Vaginal	
OncoGel™, a controlled-release depot formulation of paclitaxel in ReGel™, Protherics Salt Lake City, Inc. (Salt Lake City, UT, USA) [42,43,44]	PLGA and PEG	paclitaxel	drug dissolved or dispersed in ReGel™	provides a depot of ReGel™ for the continuous release of paclitaxel directly to the tumor and surrounding tissue for 6 weeks.	Intralesional	temperature
Thermosensitive poloxamer-hyaluronic acid-kappa-carrageenan-based hydrogel anti-adhesive agent loaded with 5-fluorouracil [4]	Poloxamer, hyaluronic acid, kappa-carrageenan	5-fluorouracil	drug dissolved in aqueous gel solution	The drug was initially burst released, the steady-state was achieved after 3–4 h, sustained release up to 3 days	Intraperitoneal	temperature
Silane-graphene dispersed cross-linked vinyl carboxymethyl chitosan and PNIPAM hydrogel [45]	Silane-graphene, vinyl carboxymethyl chitosan, PNIPAM	ciprofloxacin hydrochloride		The cumulative drug release increased rapidly in the first 7 h and remained constant after 7 h	Intranasal	temperature
Bioadhesive hydrogels for spinal cord injury [46]	Chemically modified hyaluronic acid with dopamine	Ibuprofen	dopamine chemically conjugated to hyaluronic backbone via ester or amide bonds; ibuprofen dissolved or dispersed in the aqueous pre-gel solution		Intrathecal	
Calcium-responsive composite hydrogel for acute spinal cord injury treatment [47,48]	Alginate, chitosan, and genipin (cross-linking agent)			In situ forming (rapid cross-linking) creates a cross-linking gradient, making the formation of a homogeneously cross-linked hydrogel difficult. The saturation of alginate/calcium cross-linking on the hydrogel’s surface significantly limits diffusion, resulting in slow drug release. ^3^	Intrathecal	Ca^2+^ concentration
Ozurdex^®^, an intravitreal implant [49,50]	PLGA	dexamethasone	drug embedded in gel matrix using hot-melt extrusion	Ozurdex^®^ is injected directly into the vitreous humor. The implant slowly releases dexamethasone for up to six months.	Intravitreal	
Nanoparticle-loaded ring implant placed between partially polymerized hydrogel contact lenses [51]	ethyl cellulose-nanoparticles incorporated in poly-hydroxyethyl methyl acrylate (HEMA), and poly[HEMA-co-methacrylic acid] loaded in polypropylene lens mold.	Timolol maleate	pre-formed gel soaked in drug solution	In vitro sustained drug release within the therapeutic window for 168 h and in vivo tear fluid release for more than 192 h	Ocular	
Thermosensitive chitosan/gelatin hydrogel eye drop [52]	Chitosan, gelatin, glycerol	Latanoprost	drug dissolved or dispersed in the aqueous pre-gel solution	sustained-release profile both in vitro and in vivo for 7 days	Ocular	temperature

* Methods of conjugation to incorporate small-molecule drug components into hydrogels include physical entrapment and chemical conjugation. ^1^ The hydrogel cannot be subjected to oral administration but can be administered for intravenous drug release. ^2^ Although the interaction between doxorubicin and hydrogel anionic groups can be considered as chemical conjugation, it is primarily an electrostatic attraction. In this context, doxorubicin is physically entrapped in the hydrogel system. ^3^ A previous study has shown that alginate hydrogels can be formed in the eye (in situ gelation) using simulated tears with about 0.5 mM CaCl_2_. For instance, hydrogels formed in situ within simulated tear fluid showed the slowest drug release rate [48]. Therefore, hydrogels with consistent and homogenous physical and mechanical properties during the interaction with Ca^2+^ ions in the surrounding environment are ideal for avoiding unpredictable and variable cellular behavior in vivo.

## 3. Conjugation of Biologics to Hydrogels

The typical administration of a hydrogel formulation begins with its early use in free-drug wound dressing and continues with its extensively developed complicated applications with multiple types of stimuli for controlled drug delivery and tissue engineering. For various therapeutic purposes, hydrogels and biomolecules can be incorporated through cross-linking and encapsulation techniques.

### 3.1. Nucleic Acids-Conjugated Hydrogels

Nucleic acid-based therapeutics offer great promise in disease intervention. DNA, RNA, and other oligonucleotides have been widely used in biomedical applications, specifically, hydrogel-loading of nucleic acid materials for controlled release and sustained delivery [53]. Researchers usually load these nucleotides into a hydrogel by mixing the DNA or RNA solution directly with the hydrogel precursor solution, allowing the nucleotide chains to become entrapped within the forming hydrogel network [54]. Hydrogel helps protect the DNA/RNA, maintaining their biological activity and stability, and enables retention and sustained release of encapsulated DNA/RNA. Although simply loading DNA/RNA into a hydrogel can lead to uneven distribution of encapsulated DNA/RNA and leakage of nucleotides [54,55,56,57,58], the possibility of future long-term, high-density DNA storage via hydrogels was reported. In the study, thermally responsive functionally graded hydrogels are made of a poly (NIPAM/sodium acrylate) semi-interpenetrating polymer network that is then swollen in a hyperbranched polyethylene solution and binds to DNA. The swelling of the hydrogels helps absorb DNA, and the deswelling of the hydrogels traps them; the repeat cycle of swelling-deswelling concentrates the DNA up to a DNA data density of 10^9^ GB/g [59].

While both RNA and DNA are used in hydrogel technology, RNA is usually physically encapsulated or electrostatically bound within the hydrogel network rather than polymerized into a hydrogel network. This is because most RNAs, unlike DNAs, lack the chemical properties to participate directly in polymerization reactions. In contrast, DNA strands are more commonly used to initiate or direct hydrogel formation. An example of procedure schemes for the conjugation of DNA to polymer chains is presented in Figure 2. DNA is a natural polymer composed of nucleotide monomers with a semi-flexible backbone. Its sequence-specific hybridization properties allow DNA to be programmed through base-pairing, forming 2D or 3D nanostructures [60,61,62]. Therefore, DNA–hydrogel conjugates can be described as a 3D network of DNA strands cross-linked and conjugated with functional groups [62]. The most common DNA–hydrogel conjugates include hydrogels made up entirely of DNA through hybridization or polymerization by enzymatic extension [e.g., polymerase chain reaction (PCR), hybridization chain reaction (HCR)]. In the hybridization approach, thermally stable, cross-linkable, linear, and branched DNA monomers were designed and synthesized, then hybridized and ligated with each other via DNA ligase. A core–shell spherical 3D DNA hydrogel, developed by Li et al. using HCR shows promise for synergistic cancer therapy. The hydrogel core, encapsulated within a synergistic targeting liposome membrane, is formed by polymerizing DNA using siRNA as an initiator and H1–H4 hairpins [63]. In the PCR approach, DNA nanostructures with multiple branching arms were engineered to be highly heat-resistant and used as modular primers for PCR [64,65]. In the alternative approach, long DNA strands are generated through rolling circle amplification (RCA), eventually entangling to form a gel [66]. Branched DNA hydrogels offer a versatile platform for delivering therapeutic agents enabling selective drug release in vivo, although the formation of branched DNA-based hydrogels often requires high DNA concentrations which can indeed drive up the costs.

Other conjugations of DNA and hydrogels include hydrogels containing DNA as a functional graft and hybrid DNA hydrogels with DNA as a cross-linker in polymer subunits (Table 3). An interesting example of hydrogel containing DNA as a functional graft is DNAzyme hydrogel, in which DNAzyme strands are specifically grafted onto the polymer chains to make the functional hydrogels. The system can bind to target mRNA via complementary base pairing and efficiently cleave mRNA on-site with metal ions as a cofactor due to the catalytic activity of DNAzymes, which enables cleaving RNA substrates or reacting in the presence of Mg^2+^ or Pb^2+^ ions [67,68,69,70]. Therefore, by triggering the catalytic activity of the DNAzyme, the hydrogel can release drugs in a controlled manner. DNAzymes catalyze reactions that lead to detectable signals, making these hydrogels biosensors (Table 4).

Table 3 provides a detailed classification of DNA–hydrogel conjugates based on their fabrication methods, outlining the strategies and mechanisms of these methods as they are used to synthesize DNA–hydrogel conjugates. Table 4 lists different therapeutic agents that can be co-loaded into DNA hydrogels to achieve multiple therapeutic effects. Boolean logic operations can be applied to control the gelation, degradation, and drug release of the nucleic-based-hydrogel systems based on DNA strands’ unique properties [31,71]. When the hydrogel degrades, and the drug is released (Figure 4), the DNA strands within the hydrogel can bind to DNA sequences in the environment, or, on the contrary, DNA from outside can invade and bind to a pre-hybridized sequence, displacing the DNA within the hydrogel [31]. The successful construction of DNA-based micro liquid droplet complexes that can perform programmable actions, including the Boolean logic operation, by responding to molecular inputs demonstrated that the liquid droplets are more dynamic with fluid-like internal mobility compared to injectable polymeric cross-linked hydrogels [71,72].

**Table 4 pharmaceutics-17-00897-t004:** Examples of DNA/RNA hydrogel conjugates as drug delivery systems.

	DNA/RNA Hydrogel Constituents	Drugs	Release	Administration Routes	Stimuli-Responsive
DNA–RNA hybrid hydrogel for RNA release [53]	polymerized circular DNAs of AS 1411 aptamer and GFP siRNA	GFP siRNA			
ordered structure of DNA hydrogel formed and self-assembled by polyadenine strands and cyanuric acid through hydrogen bonding [57]	polyadenine strands and cyanuric acid	DNA, DNA nanostructures, and gene-silencing nucleic acids (antisense oligonucleotides)	controlled release in a pH-responsive manner	Injection forming in situ depot	pH
Core–shell spherical 3D siRNA framework nucleic acids [63]	HCR initiated by siRNA with 4 hairpin DNA monomers, terminated by gDNA, covered with noncationic liposome membranes	doxorubicin	multiple Boolean logic gates arranged in a sequence, where the output of one gate acts as the input for the next, allow controlled release of doxorubicin and DNA in the presence of logical stimulators (glutathione, ATP, and survivin mRNA)	intravenous injection	tumor microenvironments and folate receptor overexpression
Injectable DNA supramolecular hydrogel vaccine system [73]	Y-scaffold DNA and DNA linkers assemble through non-covalent interactions	Antigens		Injection forming in situ depot	
	Immunostimulatory receptor molecules (CpG motifs) can be integrated into the DNA sequences within the hydrogel to enhance immune response.				
pH-responsive DNA hydrogel for mRNA delivery [74]	nanosphere formation from cross-linking X-shaped DNA scaffolds containing a pH-responsive i-motif sequence and DNA linkers	mRNA encoding Gluc	The hydrogel was stable under neutral pH, but at pH 4.5–5, it disintegrated and released the mRNA after being endocytosed into cells through the lysosome.	-	pH
“I-gel” enabling in situ siRNA production [75]	DNA hydrogel scaffolds incorporated with plasmid DNA encoding siRNA gene	Plasmid DNA, siRNA			
Injectable DNA–chitosan hybrid hydrogel [76]	DNA gel cross-linked with chitosan	dexamethasone	The gel degraded rapidly without coating. The release of dexamethasone corresponded with the gel degradation.	Injection	
Photothermal polydopamine-coated DNA gel [77]	siRNA cross-linked with DNA-grafted polycaprolactone, coated with polydopamine and surface PEGylation	siRNA	Slow release. siRNAs were released after 36 h. Acidic conditions induce photothermal conversion, polydopamine degradation, and siRNA release.	Intravenous injection	Light energy heat
self-assembled, dendrimer complexed RNA-hydrogel [78]	miRNAs formed complexation with polyamidoamin dendrimers	Two types of miRNAs: a miR mimic and an antagomiR	tumor suppressor miR was released first, then oncogenic miR was released	Intraperitoneal injection	
Self-assembled, layered RNA dendrimers for layer-by-layer release [79]	4 generations of RNA dendrimers	paclitaxel	RNA assembly degraded, leading to the exposure of RNA-paclitaxel linkage. Paclitaxel was released from the linkage through hydrolysis with low cytokine release.	Injection	

### 3.2. Protein- and Peptide-Conjugated Hydrogels

Therapeutic biologics such as proteins and peptides offer an opportunity for treatment that resembles natural physiological pathways, thus delivering drugs such as insulin and hormones that can address a wide range of unmet medical needs. Hydrogels provide a 3D environment to entrap and protect the bioactive molecules from degradation. However, physical entrapment with no covalent binding often leads to leakage and faster release (Figure 3).

Figure 5 depicts reaction schemes for the conjugation of bioactive peptides to polymers, with the most common first steps involving coupling reactions targeting amine (–NH_2_) or thiol (–SH) groups (1) typically resulting in self-assembled supramolecular structure or (2) followed by polymerization, resulting in multiple bioconjugates (hydrogels). Proteins are composed of one or more polypeptide chains; therefore, conjugating proteins to hydrogels can start with coupling reactions targeting amine groups (e.g., the ε-amine of lysine residues form stable amide bonds with an NHS-activated carboxylic acid group of hydrogels) [80]. Protein unfolding-chemical coupling strategy, which involves forming amide bonds, is a widely used method for creating protein-hydrogel conjugates [80,81,82]. Byeon et al. reported a reaction between the modified Z1Z2 domain-based protein and a 4-arm PEG-maleimide, enabling cross-linking. The conjugates then self-assemble to form the final hydrogel network [83]. PEG’s terminal -OH groups can be replaced with reactive groups, allowing targeted conjugation to amines, thiols, or other nucleophilic sites of biomolecules. Since PEG is non-immunogenic and can self-assemble and undergo reversible gelation, protein- and peptide-PEG conjugates remain the most common type of bioconjugates [77,82,83,84,85]. However, it is important to note that while PEG is generally a safe, well-tolerated material, PEG-specific antibodies may develop. Table 5 categorizes protein-/peptide-hydrogel conjugates into: (1) self-assembly and physical entrapment, and (2) chemical conjugation. The physical methods are simple and do not require chemical modification, resulting in reversible assemblies that allow dynamic release. On the other hand, chemical conjugation’s advantages include high specificity and creating strong bonds through fast reaction kinetics. The choice of method depends heavily on the desired properties of hydrogel conjugates.

Inspired by the common sugars and amino acid conjugates formed through the Maillard reaction in the food industry, as alternatives for synthetic polymers, polysaccharides such as alginate [47,48,86,87], chitosan [76,88,89,90,91], hyaluronic acid [83], carrageenan, and pectin have been used for biomedical applications due to their biocompatibility, natural occurrence, and ability to form stable hydrogels with various cross-linking types and varying degrees of cross-linking. Additionally, the fact that alginate can gel in the presence of divalent ions has been applied to construct a peptide-conjugated hydrogel via ion-triggered co-assembly of peptide and alginate [47,48,92]. Many negatively charged polysaccharides can form a gel upon interacting with positively charged ions, attracting water molecules, and contributing to their hydration. Interestingly, chitosan is positively charged due to its amino groups, making chitosan bind to any negatively charged surfaces including nucleic acids (e.g., siRNA [93,94], miRNA [95], DNA [76,91], etc.), peptides (e.g., RGD [95,96]), antibodies, and growth factors (e.g., TGF-β1 [97]). Chitosan is often prepared in acidic solutions, which can restrict the product design. Chitosan-derived hydrogels may also be affected by the pH of the external environment. The general disadvantage of polysaccharide–biologic conjugates is their mechanical weakness, which leads to their quick degradation and instability.

**Table 5 pharmaceutics-17-00897-t005:** Classification of proteins/peptides-hydrogel conjugates and methods for protein/peptide-hydrogel conjugates.

**Protein/Peptide-Hydrogel Conjugates**	**Self-Assembly and Physical Entrapment**	**Chemical Conjugation**
Methods	Positively charged proteins bind to negatively charged hydrogels. Hydrophobic proteins associate with hydrophobic hydrogels Heparin has a high affinity for fibroblast [98], vascular endothelial, and hepatocyte growth factors [99,100], enabling selective binding. Avidin binds to biotin (vitamin H/vitamin B7) [101]	Functional group (–NH_2_, –SH, –COOH)-based coupling (Figure 5) (e.g., collagen–alginate conjugates form via amine groups on collagen and carboxyl groups on alginate) “click” reaction [102] (e.g., azide–alkyne cycloaddition) Enzymatic ligation (e.g., transglutaminase-mediated cross-linking)

Hydrogels are the preferred material to interface with extracellular matrices. They are suitable for wound-healing applications because of their high-water content and nutritional permeability. Bioactive peptides and functional proteins can be conjugated to hydrogels to create a cell-responsive matrix that can rebuild the wounded tissues, leading to the application of hydrogels in tissue engineering and regenerative medicine [12,90,103,104,105].

### 3.3. Cell-Conjugated Hydrogels

Three-dimensional hyaluronic acid, alginate, and chitosan-based hydrogel scaffolding networks are specifically designed to mimic the structure and function of the extracellular matrix, promoting cell growth and adhesion [106]. In wound dressings, hydrogels are involved in debridement and hydration, facilitating wound healing by guiding tissue regeneration. Furthermore, active ingredients can be loaded into the hydrogel to accelerate the wound healing process or therapeutically affect the wound site. Delivering cell-based therapies to tissue defects using hydrogel-based delivery systems relies on cell encapsulation within the hydrogel network. Hydrogel-based cell therapy has gained a lot of attention in recent decades.

Hydrogels can be used to encapsulate various cell types for tissue repair, such as stem cells, islet cells, hepatocytes, and endothelial cells, providing them with an environment to grow and function [105,106,107,108,109,110,111,112]. Hydrogels containing cells and cell-derived extracellular matrices enhance cell survival, adhesion, differentiation, and proliferation. Adoptive cell therapy studies showed that immune cells, including T cells, natural killer (NK) cells, dendritic cells, or macrophages, were encapsulated within hydrogel systems for immunotherapy at tumor sites [113,114,115,116,117,118,119]. Table 6 summarizes a range of cell–hydrogel conjugates. Methods of conjugation for cell–hydrogel conjugates include: (1) physical entrapment (encapsulation), in which cells are encapsulated within the hydrogel or seeded onto the scaffold (3D cell culture); (2) chemical conjugation, which stems from protein unfolding-chemical coupling strategy forming covalent bonds between hydrogel component and functional groups on cell membrane (e.g., amines, thiols, etc.); and (3) conjugation through bio affinity between ligand-receptor or protein–protein interactions [101]. We observe that both natural (hyaluronic acid, collagen, chitosan, and alginate) and synthetic polymers were employed to construct hydrogels. Natural polymers mimic the nature of the extracellular matrix and support cell attachment and growth. However, synthetic polymers offer tunable mechanical properties and controlled degradation for controlled release or to promote cell interactions. Recently developed advanced hydrogel systems incorporate modified biomolecules (e.g., cholesterol-modified miRNA mimics) [108], functional peptides, or combine 2 or more polymers to harness both the biological advantages of cell therapy and the benefits of robust mechanical properties of the hydrogel, ensuring that cells are supported by a bioactive matrix while maintaining precise control over the scaffold’s physical properties and release kinetics. A treatment that combines cells and therapeutic agents is the next generation regenerative and smart therapeutics. For example, pH sensitive, hydrogel-structured phenylboronate ester was created from phenylboronic-modified chitosan, PVA, and benzaldehyde-capped PEG, was then used to encapsulate insulin and L929 fibroblasts, enabling a combination therapy including insulin delivery and cellular support. In rat model, wound closure was accelerated with steady insulin release and enhanced neovascularization. This is a representation for smart hydrogel wound dressing system that conjugates both proteins and cells, resulting in collagen deposition, improved angiogenesis specifically in high-glucose, low pH diabetic wounds [85]. The release profile of insulin and fibroblasts from the dual-responsive hydrogel triggered by glucose and pH followed a multi-phase pattern with burst release of insulin, followed by stimulus-triggered release and sustained release of insulin. Interestingly, the cells remain encapsulated in the hydrogel network until the hydrogel erodes, causing them to migrate out, thus secreting their cytokines and growth factors for tissue regeneration.

Even though for hydrogel-loaded therapeutic agents, especially small-molecule drugs, “burst release” is sometimes useful for a faster release profile, it is often problematic for biologics, causing a loss of bioactivity and a high immune risk. It is noteworthy to understand that a “burst release” of cells from a cell-conjugated hydrogel system can be disastrous, unlike with drugs, because cells are often intended for tissue regeneration or interact with host cells at the site of action.

**Table 6 pharmaceutics-17-00897-t006:** Examples of cell–hydrogel conjugates.

Cell–Hydrogel Conjugates	Hydrogel Constituents	Cell Type	Method of Conjugations	Specifications	**Administration Routes**	**Stimuli-Responsive**
Multi-layer hydrogel-loaded mesenchymal stem cells for cartilage repair [107]	PEG di-methacrylate, methacrylate chondroitin sulfate	Mesenchymal stem cells	Physical entrapment (encapsulation)			
Injectable stem cell-based hydrogel delivering miRNA for cartilage repair [108]	a cholesterol-modified miRNA mimic, bone marrow–homing and stem cell–homing peptides assembly	Synovium-derived mesenchymal stem cells		Sustained delivery of miRNA and recruitment of stem cells for cartilage repair	Intra-articular injection	
Injectable TGF-β1 conjugated chitosan, collagen hydrogel for articular cartilage regeneration [109]	TGF-β1 growth and chondrogenic factors conjugated chitosan, collagen	Mesenchymal stem cells	Mesenchymal stem cells embedded in the gel matrix (physical entrapment); TGF-β1 growth and chondrogenic factors are chemically linked to chitosan through covalent bonds (covalent conjugation)	The delivery of TGF-β1 growth factor and collagen promoted cellular aggregation and deposition of cartilaginous extracellular matrix	Injection	
Alginate-collagen encapsulation of human pancreatic islet cells for transplantation [110]	Alginate, collagen ^1^	Islet cells	cells suspended in gel solution (Physical entrapment)	Collagen improves islet cell survival [110]. Although, alginate is often used for islet cell encapsulation. Alginate cannot support islet cell survival [111].		
3D hydrogel tissue model for liver tissue engineering [105,112]	Gelatin methacryloyl	Hepatocytes, fibroblasts		The tissue model maintained over 90% cell viability.		
T-cell stimulating hydrogel matrix for immunotherapy [113]	Thiolated hyaluronic acid cross-linked with PEG diacrylate	T cells		T cell activation		T cell stimulatory signals
T-cell-responsive hydrogels for in situ T-cell expansion and enhanced antitumor efficacy [114]	Dynabeads, PEG, alginate	T cells	Dynabeads embedded in the gel solution; specific bio-affinity between Dynabeads (T-cell-specific antibody-coated beads) and T-cells; T-cells entrapped in the gel matrix	Enabled controlled release of T cell activation and facilitated T cell expansion as well es enhanced antitumor efficacy.	Injectable	T cell stimulatory signals
Hydrogel-releasing CAR-T cells and anti-PDL1-conjugated platelets [115]	Acrylated hyaluronic acid	CAR T cells, cytokine interleukin-15 (IL-15), platelets conjugated with the checkpoint inhibitor programmed death-ligand 1 ^2^	physical entrapment	sustained release of CAR-T cells targeting the human chondroitin sulfate proteoglycan 4 and IL-15	Intraoperative patch	cytokine IL-15
Biopolymeric implant for the delivery of T-cell therapy [116]	Alginate, collagen	T cells		Target delivery to tumor site for the release, expansion, and dispersion of T cells	Implantation	
Injectable hydrogels for controlled co-delivery of CAR-T cells and stimulatory cytokines [117]	HPMC, RGD-PEG-PLA nanoparticles ^3^	CAR-T cells, cytokine IL-15		IL-15 slowly diffused and released, activated T cells. CAR-T cells were continuously released over 8 days from 2 formulations with >85% viability, enabling 4.5-fold enhancement CAR-T cell exposure compared to standard bolus administration.	Injection	cytokine IL-15
Injectable thermosensitive hydrogels for controlled delivery of NK cells against solid tumor [118]	Hydroxyapatite-modified chitosan	NK cells		NK cell stimulation and release for tumor immunotherapy.	Injection	temperature
Localized in situ gelling PEG-based hydrogel for multiple sclerosis [119]	PEG	Dendritic cells treated with IL-10	PEG attached to dendritic cells (bio affinity-based conjugation)	Cells were delivered locally, altered the injection site recruited, increased endogenous immune cell profile within 2 days.	Injection	

^1.^ collagen conjugates to alginate oligomers via chemical conjugation. ^2.^ platelets can be chemically conjugated with IL-1 blockers using standard functional group coupling. ^3.^ RGD peptides are chemically linked to PEG-PLA via –NHS-ester coupling.

## 4. Conclusions

Hydrogel conjugation offers a versatile platform for advanced drug delivery. By linking therapeutic agents—ranging from chemicals to biologicals—to a 3D, hydrophilic polymer network, hydrogel conjugates represent transformative, advanced delivery systems with controlled release, enhanced active ingredient/drug stability, and sustained therapeutic effects for wound healing and targeted drug delivery.

This review highlights the utility and effectiveness of hydrogel conjugates involving small molecules and macromolecules and explores their conjugation methods, emphasizing the potential of conjugating diverse therapeutic agents with hydrogels. Emerging approaches focus on expanding the toolkit available for hydrogel conjugation, including innovative physical assemblies, chemical binding motifs, stimuli-responsive linkages, and functional group modifications. These advances have led to the development of various dosage forms for various administration routes. Although hydrogels are frequently designed to be injectable, injectable hydrogels are delivered as liquid precursors that are minimally invasive and can undergo gelation triggered by stimuli. Injectability offers 3D-bioprintable materials and fillers for complex tissue architectures. A step further than tunability and injectability is the potential for “self-healing hydrogels”—hydrogels that can automatically reform themselves after being damaged. The reformation can be achieved through reactions or interactions that allow polymeric or co-polymeric chains to reform their bonds within the hydrogel.

By providing a description and discussion of several bio-conjugated hydrogel systems with their applications, as well as clinical translations, this review underscores the hydrogel conjugates that are made up of combination polymers and combined therapy, resulting in better environments for controlled drug delivery and regenerative medicine. Further research should focus on optimizing hydrogel conjugates and refining the formulations with in vivo outcomes, which are essential and clinically relevant. There is also a gap in the literature as pertains to scale-up design, development, and scalable production evaluation for hydrogels, as well as an absence of short-term/long-term storage stability assessments. Moreover, drug-conjugated hydrogels face significant regulatory challenges, and their economic feasibility is another critical factor influencing their adoption.

## Figures and Tables

**Figure 1 pharmaceutics-17-00897-f001:**
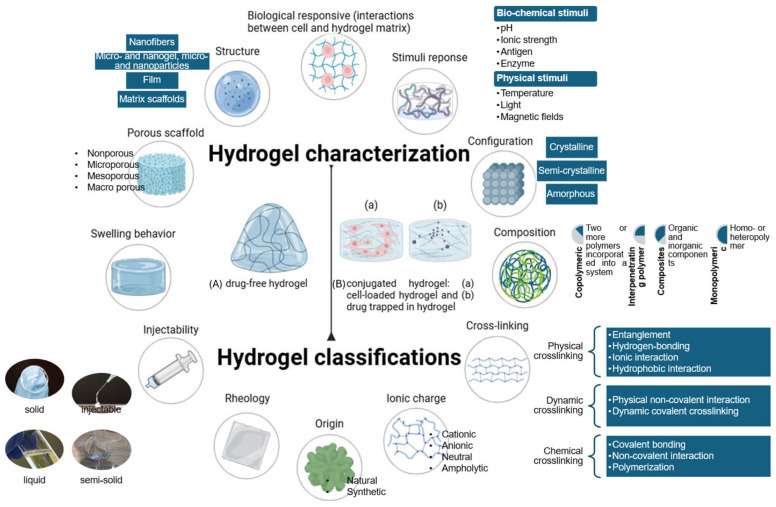
Comprehensive overview of hydrogel characterization and classification by measurable factors, characteristics, and physicochemical properties. The development of hydrogel systems for biomedical applications relies on hydrogel material qualities such as origin (natural or synthetic), composition (homo-, hetero-, or co-polymer), ionic charge (cationic, anionic, neutral, or ampholytic), configuration (crystalline or amorphous), rheology, injectability, swelling behavior and degradation rate, porosity, structure, cross-link type (physical, chemical, or dynamic), and cross-linking density. Biologically responsive interactions between cells and the hydrogel matrix are influenced by stimuli such as pH, ionic strength, antigen, enzyme, physical strain, and magnetic fields. The factors, characteristics, and properties are illustrated in a circular layout to emphasize their interconnectedness and interrelatedness. In the center, biomedical hydrogel products can be classified into (A) drug-free hydrogel and (B) conjugated hydrogel, including (a) cell-loaded hydrogel and (b) drug molecule-loaded hydrogels.

**Figure 2 pharmaceutics-17-00897-f002:**
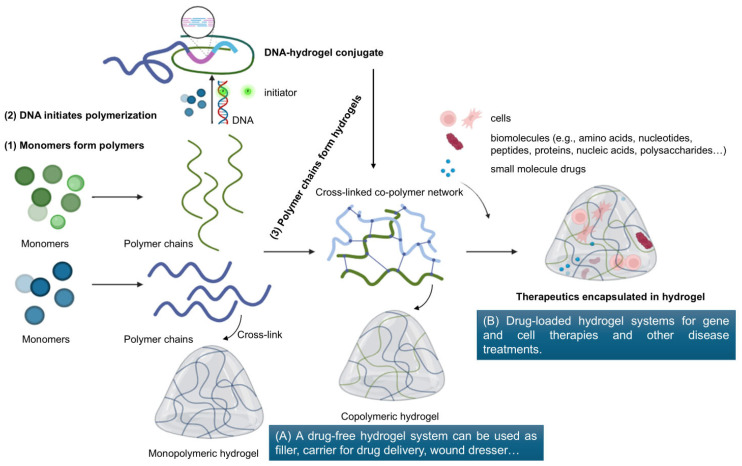
Schematic images for the steps involved in conjugated hydrogel synthesis, illustrating hydrogel products for wound dressings and therapeutic delivery. The formation of hydrogels is depicted, starting with (1) monomers, (2) DNA, or (3) polymer chains.

**Figure 3 pharmaceutics-17-00897-f003:**
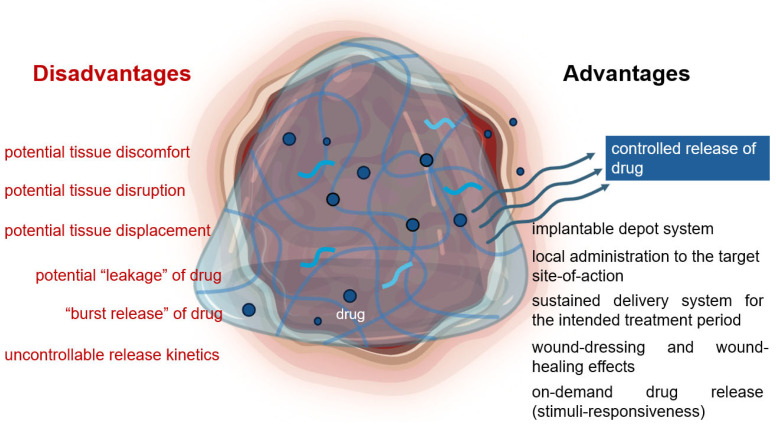
Illustration of hydrogel enabling a localized, controlled, and sustained delivery of drug. The hydrogel wound dressing is implanted at the target site. It suppresses inflammatory reactions and can adhere at the target site to deliver drugs for the intended treatment period. However, some disadvantages are potential tissue discomfort and even tissue damage. Maintaining the hydrogel position and controlling hydrogel erosion is difficult, which may lead to displacement or drug leakage. Incorporating hydrophobic drugs into hydrogel remains a challenge. Meanwhile, hydrophilic drugs experience “burst release” in “water-swelling” hydrogels.

**Figure 4 pharmaceutics-17-00897-f004:**
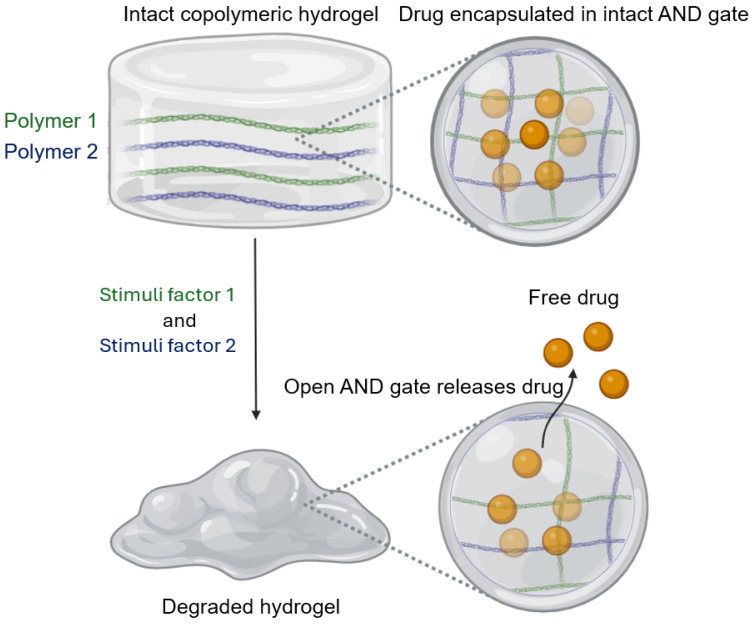
Illustration of an AND gate enabling drug release from a hydrogel. The drug is released only if both polymer 1 and polymer 2 are triggered by stimuli factors 1 and 2.

**Figure 5 pharmaceutics-17-00897-f005:**
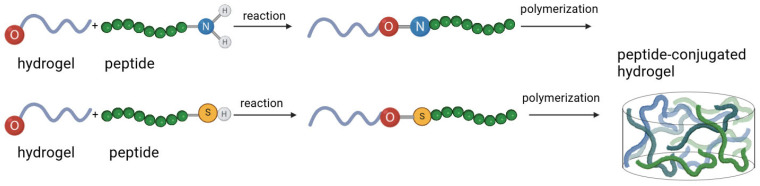
Schemes for the conjugation and polymerization for linking proteins (polypeptides) and peptides to hydrogels. The conjugation typically starts with coupling reactions targeting amine (–NH_2_) or thiol (–SH) groups, followed by polymerization, forming multiple conjugates, resulting in self-assembled supramolecular structures.

**Table 1 pharmaceutics-17-00897-t001:** Classification of hydrogel–drug conjugation based on drug component.

**Hydrogel –drug conjugation**	**Drug component**			
	**Hydrogel component**	**Drug type**		
		**Small molecules**		chemical compounds, usually <1000 Da
		**Biologics**	**Monomers**	single molecule units (e.g., amino acids, nucleotides, etc.)
			**Polymers**	large biomolecules including peptides, proteins, nucleic acids (e.g., DNA, RNA), and polysaccharides.
			**Cells**	complex systems made up of polymeric biomolecules.
		**Other biotechnological drug delivery systems (carriers)**		micelles, liposomes, dendrimers, micro- and nanoparticles…

**Table 3 pharmaceutics-17-00897-t003:** Classification of DNA–hydrogel conjugates and methods for DNA–hydrogel conjugates.

DNA–Hydrogel Conjugates	Hydrogels Made Entirely of DNA		Hydrogels Containing DNA as a Functional Graft		Hydrogels with DNA as a Cross-Linker in Polymer Subunits
Methods	Self-assembly				
	Enzymatic reactions	Ligation			
		Hybridization		PCR	
				RCA: amplifying DNA using a circular DNA template	
				Multi-primed chain amplification (MCA) and/or multiple displacement amplification (MDA): significantly amplifying small amounts of DNA, allowing amplification from minimal starting sequences	
				Clamped-hybridization chain reaction (C-HCR): Two DNA hairpins self-assembled into a larger structure through hybridization, with an initiator strand “clamp” to trigger the assembly process, forming a hydrogel.	
	Chemical cross-linkers	ethylene glycol diglycidyl ether (EGDE) and polyethylene glycol diglycidyl ether (PEGDE)			
		tetramethyl ethylenediamine (TEMED)			

## Data Availability

Not applicable.

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
