# Peer review of "Hydrogel Conjugation: Engineering of Hydrogels for Drug Delivery"

_pharmaceutics, 2025, doi:10.3390/pharmaceutics17070897_

Round 1
Reviewer 1 Report
Comments and Suggestions for Authors
This article on hydrogel conjugation for drug delivery systems provides a thorough overview of current methodologies, applications, and challenges. The manuscript is well-structured and addresses a topic of significant interest in pharmaceutical sciences. However, several issues require attention to enhance clarity, accuracy, and scientific rigor. Below are specific recommendations for improvement.
- The abstract should explicitly mention the novelty of this review compared to existing literature. While it outlines objectives, it lacks a concise summary of key findings or advancements in hydrogel conjugation strategies.Avoid vague phrases like “various biomedical applications” and replace them with specific examples.
- The format of the abstract does not meet the requirements of a regular review paper. Please revise it.
- The format of Table 1 is different from other tables. In addition, for the convenience of readers, the font in the table should be left aligned.
- The statement “DNA is technically a natural, semi-flexible polymer” is misleading.
- The captions for Figures 1 and 2 are insufficient.
- The format of references 11-12 is incorrect, and references 109-117 are missing.
Reviewer 2 Report
Comments and Suggestions for Authors
- The expression of this review is not fluent. And the figures contain much content, but the relationship among these elements is not clear.
- The review introduced small-molecule drugs, nucleic acids and proteins. However, these three kinds of drugs have totally different physicochemical properties. Thus, the conjugation mechanism should be analyzed carefully respectively.
- The application of various detailed samples as small-molecule drugs, nucleic acids and proteins should be discussed.
Reviewer 3 Report
Comments and Suggestions for Authors
This review highlights the hydrogel conjugation process involving small molecules and macromolecules and their conjugation methods. Subsequently, the description and discussion of several bio-conjugated hydrogel systems and their potential for clinical translation are provided. While the article presents innovative concepts, the logical flow could be strengthened, and the figures and tables require standardization to meet publication requirements. We recommend addressing the specific points outlined below.
- Please add subheadings to the article to improve the logic of the article.
- Table 1 is not clear. Please provide a clearer table, e.g. by adding supporting lines.
- Figure 2 would benefit from refinement to improve clarity and focus. Please consider: Streamlining the text elements, enhancing the visual hierarchy, more precise labeling to direct reader attention to better communicate the key concepts.
- If the figures in the review are from references, please ensure that approval for their use has been obtained and that the source is indicated in the article.
- Table 2 and 3 requires formatting adjustments to meet journal standards. Please: adjust column widths for proper alignment, reduce text size to improve readability and ensure consistent styling throughout the table.
- Please put unnecessary forms in to supporting documentation.
- Please increase the logic of the second part of the article, such as adding subheadings.
- Are there any key linkages between parts 2 and 3? Are there other types of categories besides these two parts? Please add relevance information.
- The logic of the conclusion is confusing.
- Correct grammatical errors.
Reviewer 4 Report
Comments and Suggestions for Authors
The reviewed manuscript, ''Hydrogel conjugation: Engineering of hydrogels for drug delivery'' provides an overview of the engineering of conjugated hydrogels and covers a wide range of topics. However, there are several areas where the paper could be improved to improve its scientific readability.
I suggest the following to the authors:
- The manuscript lacks a clear distinction between the already reported work on hydrogels as drug delivery and controlled release systems and the new contributions made by this review. Please emphasize the unique aspects more explicitly.
- Table 1 is not convincing. It should be rewritten.
- Section 2 presents general characteristics of hydrogels as drug delivery systems and is not very well organized. Here, the specific characteristics of ''Conjugation of small molecules to hydrogel'' should be highlighted, as indicated in the title.
- Table 2 should include a column specifying the method of conjugation of drug molecules to the presented hydrogels.
- Two types of reactions (1 and 2) are specified in R 251. These indices are not found in the legend of Figure 5.
- In subsection 3.2 titled ''Proteins, and peptides-conjugated hydrogels'' only peptides are discussed. Please add proteins and discuss the method of their conjugation to hydrogels, arguing with references.
- Table 4 is chaotic. Please make a better classification of protein/peptide conjugated hydrogels, with references.
- Subsection 3.3. Cell-conjugated hydrogels should be redone. Please pay more attention and describe the methods of conjugation of hydrogels with different types of cells.
- The manuscript summarizes many studies, but lacks a more in-depth critical assessment of their limitations and challenges regarding the conjugation of hydrogels in drug delivery.
- A discussion of the biodegradability and long-term stability of drug-carrying hydrogels is missing. Please elaborate on the degradation mechanisms, rates, and potential side effects of these conjugated hydrogels as drug delivery systems.
- It would be useful to add a section dedicated to the regulatory challenges, manufacturing scalability, and cost-effectiveness of drug-conjugated hydrogels to increase the relevance of this review.
- The following recent reference could be cited: 10.5772/intechopen.1004826
- If possible, add a graphical abstract.
Round 2
Reviewer 2 Report
Comments and Suggestions for Authors
The subtitles are suggested to cancel in Section Introduction.
Author Response
Dear Reviewer,
Thank you for your comments. We have removed the subsection titles of the Introduction per your suggestion.
Reviewer 3 Report
Comments and Suggestions for Authors
In the revised manuscript, the author fully revised the manuscript according to the review comments. The author also made a point to point positive response to the comments of all reviewers. I have no further comments.
Author Response
Thank you for your time and effort in reviewing my work. Your thoughtful comments and constructive suggestions were invaluable in improving the manuscript's quality.
Reviewer 4 Report
Comments and Suggestions for Authors
Following the revision, the authors made numerous changes to the manuscript with the title "Hydrogel conjugation: Engineering of hydrogels for drug delivery" and thus improved the quality of the presented work. All my concern was addressed.
Author Response
I would like to sincerely thank you for reviewing my work. Your comments and suggestions were invaluable in improving the quality of the manuscript.